# Influence of Different Harvesting Stages and Cereals–Legume Mixture on Forage Biomass Yield, Nutritional Compositions, and Quality under Loess Plateau Region

**DOI:** 10.3390/plants11202801

**Published:** 2022-10-21

**Authors:** Poe Thinzar Bo, Yinping Bai, Yongli Dong, Hongxia Shi, Maw Ni Soe Htet, Hamz Ali Samoon, Ruifang Zhang, Sikander Khan Tanveer, Jiangbo Hai

**Affiliations:** 1State Key Laboratory of Crop Cultivation and Farming System in Northwestern Loess Plateau, College of Agronomy, Northwest A & F University, Yangling, Xianyang 712100, China; 2School of Life Science and Engineering, Southwest University of Science and Technology, Mianyang 621010, China; 3State Key Laboratory of Crop Stress Biology for Arid Areas, College of Agronomy, Northwest A & F University, Yangling, Xianyang 712100, China; 4PARC-Water and Agricultural Waste Management Institute, Tando Jam 70060, Pakistan; 5National Agriculture Research Center, Islamabad 45720, Pakistan

**Keywords:** mixed cropping, harvesting stage, cereals, legume, biomass yield, nutritional compositions, forage quality index

## Abstract

One of the main problems in the animal industries currently is the constant provision of forage in sufficient amounts with acceptable nutritional content for large and small ruminants, as livestock is a significant source of income for rural people in the Loess Plateau region. Cereals and legumes are essential forage crops because of their nutritional significance, particularly the protein concentration in legumes and the fiber content in cereals. Therefore, combining cereal and legume crops may be a practical solution to the problems of inadequate forage nutrition, an insufficient amount of forage, unsustainable agricultural methods, and declining soil fertility. The current study predicts that mixed cropping of cereals and legumes at the harvesting stage of the soft dough stage and maturity stage based on the cereal growth stage will have different effects on forage biomass output, forage quality index, and nutritional value of the forage. In this study, wheat (*Triticum aestivum*) and ryegrass (*Lolium multiflorum*) are used as cereal crops and pea (*Pisum sativum*), and alfalfa (*Medicago sativa*) are used as legume crops. Three sample replicates and a split-plot design with a randomized block design are used. The study is conducted in the 2020–2021 and 2021–2022 cropping seasons. The experimental results show that cereal–legume mixed cropping, particularly the cereal–alfalfa combination, has a positive impact on the biomass yield and nutritional composition of the forage. However, adding peas to cereal has a negative impact on biomass yield, nutritional composition, mineral composition, and forage quality index. Among the treatments, ryegrass–alfalfa mixed cropping was shown to have higher values of WSC%, CP%, EE%, CF%, and ash% in both growing seasons. The values are WSC (15.82%), CP (10.78%), EE (2.30%), CF (32.06%), and ash (10.68%) for the 2020–2021 cropping seasons and WSC (15.03%), CP (11.68%), EE (3.30%), CF (32.92%), and ash (11.07%) for the 2021–2022 cropping seasons, respectively. On the other hand, the current study finds that cereal–alfalfa mixed cropping had a detrimental impact on NDF and ADF concentrations. All nutritional indices, including CP, WSC, EE, CF, ash, NDF, and ADF, have favorable correlations with one another. Furthermore, in both growing seasons, RA, ryegrass–alfalfa mixed cropping, has higher mineral compositions and forage quality indicators. Furthermore, harvesting times have a significant impact on the fresh biomass yield, dry matter yield, nutritional compositions, mineral compositions, and forage quality parameters (*p* < 0.001), with the highest values being shown when harvesting at the soft dough stage. The current study concludes that, based on chemical composition and quality analysis, the soft dough stage is the greatest harvesting period, and that the cereal–alfalfa mixed cropping is the most preferable due to its maximized quality forage production and nutritional content in livestock feedstuff in the Loess Plateau region.

## 1. Introduction

Livestock farming is an important source of income for rural people in developing countries, contributing to livelihoods through the production of meat, milk, leather, and wool. Between 1993 and 2020, demand for livestock products doubled globally, with meat and milk products expanding at annual rates of 2.7 and 3.2%, respectively, in developing countries [1]. Moreover, the demand for cattle products is rising quickly, particularly in China. Governmental initiatives seeking to double small ruminant numbers in some locations are being supported on the Loess Plateau of China to increase rural incomes [2]. Currently, China’s livestock production is primarily based on feeding collected crop residues (primarily maize and wheat) and lucerne (*Medicago sativa*), but forage shortages and their poor quality, especially during winter and early spring, restrict livestock productivity [3].

When considering the above situation, the continuous supply of forage in adequate quantities with acceptable nutritional content for large and small ruminants is one of the major issues in the animal industries of China. Moreover, forage production has further suffered from the reduction of the agricultural land area brought on by human settlements, farmers’ interest in high-revenue cash crops, and the consistently changing climate [4]. The slower forage crop growth throughout the winter is also responsible for the winter’s shortage of green fodder. Animals are typically fed nutritionally deficient dry grain stalks by farmers in this situation [5]. Therefore, it is important to concentrate on preventing production issues, creating high-quality fodder, and finding ways to improve production techniques such as cropping systems. In forage production, cereals and legumes are essential forage crops because of their nutritional significance, particularly the protein concentration in legumes and the fiber content in cereals [6]. However, fodder derived solely from cereals has poor forage quality because it contains too little protein. In order to raise the protein content of the feed, it is important to prepare forage by combining a cereal crop with a legume crop. The production and quality of forage are increased when certain annual legumes are grown in combination with cereals [7]. Furthermore, combining cereal and legume crops in one field may be a practical solution to the problems of inadequate nutrition, unsustainable agricultural methods, and declining soil fertility. In addition to collecting nutrients from the soil solution, cereals also gain from N fixed by legumes through root admixture and N supplied by legumes in the soil [8]. A resurgence in interest in these systems has already been realized in industrialized countries due to a growing understanding of the importance of creating sustainable agricultural systems for the production of grain or pasture [9]. Furthermore, the benefits of intercrops and species interactions have been evaluated using a variety of indices by some researchers [10]. In contrast to cereal monoculture, it has been demonstrated that the inclusion of legumes in combination with grasses enhances the quality of the entire forage biomass, particularly the protein content, and boosts biodiversity [11].

Additionally, a lot of researchers have assessed the benefits of combining leguminous species that are high in protein with cereals or grasses for forage production [12]. The yield of fresh biomass and dry matter yield are crucial indicators of the forage’s quality. As well, animal growth rates, reproductive success, and behavior are significantly influenced by the nutritional components of forage, including fiber, lignin, minerals, and protein. Many researchers suggest that cereal–legume intercropping can increase yields while also improving forage quality and nutritional composition [13]. Soe Htet et al. analyzed the higher fresh biomass yield and dry matter yield in the cereal–legume cropping combination in terms of a mixture of maize and common bean [14]. As well, the previous study determined that cereal (wheat, ryegrass) and legume (alfalfa) mixed cropping showed increased production of dry matter yield, fresh biomass yield, and crude protein yield [15]. Intercropping cereal (wheat) with leguminous crops to provide fodder could also give wheat the desired results. When clover and wheat were intercropped, neutral detergent fiber (NDF) and acid detergent fiber (ADF) levels increased, according to the analysis by Contreras et al. [16]. Additionally, Kim et al. came to the conclusion that Italian ryegrass and fodder pea mixed cropping contributed to better crude protein and dry matter content, as well as improved cultivation stability [17]. Furthermore, some studies revealed that cereal–legume mixed cropping has an impact on the forage quality parameters such as relative feed value (RFV), relative forage quality (RFQ), total digestible nutrient (TDN), dry matter intake (DMI), digestible dry matter (DDM), and quality index (QI). The relative feed value index (RFV) assesses cool-season legumes, grasses, and combinations based on the possible digestible dry matter intake. RFQ is an improved version of RFV. A thorough evaluation of the quality of feed is provided by the quality index (QI). The voluntary intake of available energy is estimated, and fiber amount and digestibility are also measured. The intake component is represented by DMI as a percentage of BW, as in RFV, and the accessible energy component is represented by TDN (percent of DM), as in QI [18]. Soe Htet et al. evaluated the recommended value of forage quality parameters in 50:50 ratios of maize–common bean combinations, which means the combination of cereal–legume can impact on the forage quality index [14]. In comparison to oats and barley monoculture, pea-cereal combinations produced higher CP content, TDN (total digestible nutrient), and RFV (relative feed value) values, according to research by Iqbal et al. [19].

Additionally, one of the other most important factors that affect the quality of forage feed is the time of harvest. According to Kim [17], harvesting at the right time is crucial for obtaining the maximum nutritional value as well as lowering the danger of mold infection on the farm. Different harvesting times may compromise the quality of fodder, notably the forage nutritional value [11]. On the other hand, according to the study of cereal–legume intercrops, Bacchi et al. [20] concluded that the choice of mixture and harvest time depends on the needs and target use pursued by various stakeholders. For example, if they need to produce green fodder (early harvest) with a higher nutritional value, ryegrass, and triticale in intercrops should be used. If they need silage fodder (late harvest), barley, and triticale in pure culture or in mixtures with vetch or pea. Forage crops that are harvested in the vegetative stage have low production and fiber levels, but crops harvested in the reproductive stage have the highest digestibility at the dry matter and nutritional composition levels [21]. In the study of cereal–alfalfa mixed cropping, prior investigations determined that harvesting during the flowering stage is advised for all treatments to obtain remarkable biomass production and chemical compositions [15].

It was hypothesized that the forage output of companion crops and the nutritional value of fresh fodder might be affected differently by mixed cropping of cereals (wheat and ryegrass) and legumes (pea and alfalfa) in the Loess Plateau region. The purpose of the current study is to establish the effects of cereal–legume mixed cropping on the forage biomass output, forage quality index, and nutritional value of the forage. In addition, the goal of this study was to determine the best harvesting times based on the stage of cereal growth in order to increase biomass output and nutritional benefits in fodder used for productive animal feeding in the Loess Plateau region.

## 2. Material and Methods

### 2.1. Experimental Site, Treatments, and Design

The research was carried out at the Doukou Wheat and Maize Demonstration Research Station (108°52′ E, 34°36′ N) of Northwest A&F University, Shaanxi province, P.R China, during the winter wheat cropping seasons of 2020–2021 and 2021–2022, respectively. The experimental site has a semi-humid climate. Meteorological data, recorded by Campbell scientific system, are shown in Table 1. The soil is used to classify as Earth Cumuli Orthic anthrosol. The method outlined by Piper [22] was followed when analyzing the chemical composition of the soil. Using a soil auger, soil samples were randomly taken at a depth of 0–40 cm from six randomly selected locations throughout the experimental trials. The chemical characteristics of the soil sample were assessed after air drying, grinding, and sieving. The average chemical and physical properties of the soil for both crop seasons are described in Table 2.

A randomized block experimental design with 9 treatments (W1: sole cropping of the wheat variety Xiaoyan 17, W2: monocropping of the wheat variety Baomai 9, R: sole cropping of ryegrass, W1P: mixed cropping of the wheat variety Xiaoyan 17 and pea, W2P: mixed cropping of the wheat variety Baomai 9 and pea, RP: annual ryegrass and pea mixing, W1A: mixed cropping of wheat cultivar Xiaoyan 17 and alfalfa, W2A: wheat cultivar Baomai 9 and alfalfa mixed cropping, RA: annual ryegrass and alfalfa mixing). Each treatment has three replications (Figure 1). Each plot was split for different harvesting times. Each trial measured 7 m × 3 m, with rows spaced 25 cm apart. All trials used a seed drilling system. For the study, frost-resistant cultivars of pea (P) and alfalfa (A) were used as legume crops, annual ryegrass (R), and two commercial hybrid winter wheat varieties (Xiaoyan 17, W1 and Baomai 9, W2, which have high tillering capacity and frost resistance) were applied as cereal crops. The seeding rate for cereals was 240 kg ha^−1^ and 24 kg ha^−1^ were used for legumes. The local seed ratio of 10:1 for mixed-cropping of cereals and legumes served as the basis for setting the seed ratios for mixed-cropping. All cultivars were planted on 8 October 2020, and 29 October 2021, during the growing seasons.

### 2.2. Crop Cultivation and Sample Preparation

For the purpose of preparing the seedbed, plowing and harrowing were performed at a depth of roughly 20–30 cm. Using a manual seed spreader (model AM-001100, Acme Agro-Tech Co., Ltd., Hubei, China), seeds were sown along the furrow line. Cereals and legumes were planted separately along lines that were 25 cm apart in mixed cropping. Winter wheat was grown on the field prior to the first cropping season (2020–2021); following that, the field was fallowed for 4 months. The same field was used during the second cropping season (2021–2022). Based on the results of the soil test, basal fertilizer in the form of (N-P_2_O_5_-K_2_0:24-15-5) was applied evenly throughout the field before sowing at a rate of 576 kg ha^−1^. All plots in both crop seasons established uniform cultural practices. At the tillering stage and at the stem elongation stage of cereal crops, the plots received irrigation twice in both cropping seasons. Throughout the cropping seasons, there were no infections with pests or diseases. Insecticides and herbicides were not applied. On all plots, uniform manual weeding was performed continuously.

Each cultivar was harvested in accordance with the developmental stage of cereals, in which cereals were the soft dough stage (SDS) and maturity stage (MS); for peas, it was the full pod stage and maturity stage; for alfalfa, it was the late bud stage and flowering stage. Manual shears were used to cut 1 m^2^ of each trial down to the ground. After that, the fresh sample was dried until it reached a constant weight, at which point the fresh sample’s fresh biomass yield was noted. Fresh biomass yield was converted to t ha^−1^ in value. About 300 g of fresh sample was collected from each plot, which was then chopped into 2–4 cm lengths using a power chaff cutter (JB 400, Surat, India). The samples were then dried at 65 °C for 48 h to determine the dry matter yield and percentage of dry matter. The t ha^−1^ was used to express the dry matter yield. In order to determine the quality parameters, the dried samples were ground with a grinder (FW, interior-1 Taiwan, Tianjin Xinbode Instrument Co., Ltd., Tianjin, China) and passed through a 1 mm sieve (BL-earth-soil sieve, Shanghai Baolan Experimental Instruments Manufacturing Co., Ltd., Shanghai, China).

### 2.3. Forage Quality Assessment

A Kjeldahl Analyzer (Hanon Shandong Scientific Instruments Co., Ltd., Jinan, China) was used to measure the nitrogen concentration. After that, the nitrogen concentration was multiplied by 6.25 to alter the crude protein percent (CP%) [23]. The anthrone reaction rate essay was used to calculate the water-soluble carbohydrate (WSC) [24]. A Soxhlet Extractor (Hanon Shandong Scientific Instruments Co., Ltd., Jinan, China) was applied to evaluate ether extract (EE) using the Soxhlet extraction procedure [23]. To determine the ash content, the ground samples were burned at 550 °C for three hours in a 12 L stainless steel ceramic muffle furnace (Faithful Instruments Co., Ltd., Changzhou, China) [25]. Using the ANKOM 200 fiber analyzer (ANKOM Technology, Macedon, NY, USA), crude fiber (CF), NDF, and ADF were assessed according to the method of Van Soest [26]. Mineral concentrations (Ca, Na, K, P, and Mg) were evaluated with the AOAC method [23].

Furthermore, dry matter intake (DMI), digestible dry matter (DDM), total digestible nutrient (TDN), relative forage value (RFV), relative forage quality (RFQ), and quality index (QI) were assessed by using the following formulas [27]:Dry matter intake, DMI (% of BW) = 120/(NDF, % of DM);Digestible dry matter, DDM (% of DM) = 88.9 − 0.779 × ADF (% of DM);Total digestible nutrient, TDN = 111.8 − (0.95 × % CP) − (0.36 × % ADF) − (0.7 × % NDF);Relative feed value, RFV = [(120/NDF) × (88.9 − 0.779 × ADF)]/1.29;Relative forage quality, RFQ = (DMI, % of BW) × (TDN, % of DM)/1.23;Quality index, QI = 0.0125 × RFQ + 0.097.

### 2.4. Data Analysis

The analyses of variance (a balanced one-way ANOVA) were applied to determine the treatment means significance on SPSS version 21.00 (IBM Co., Chicago, IL, USA). The least significant difference (LSD) was used for the comparison of each treatment at *p* ≤ 0.05 in Duncan’s multiple range test. Graphs were made using OriginPro 8.5 (OriginLab Corporation, Northampton, MA, USA).

## 3. Results and Discussion

### 3.1. Analysis of Variance, Fresh Biomass Yield, and Dry Matter Yield

In Table 3a,b, the fresh biomass yield, dry matter yield, nutritional compositions, mineral compositions, and forage quality parameters were significantly affected by treatment and harvesting times, whereas there was no impact of treatment on WSC content and dry matter yield, Na, and P were not affected by harvesting time. Furthermore, the year has a slight effect on NDF, DMI, DDM, TDN, RFQ, and QI and a strong effect on Na. In addition, there were no significant differences in the interaction of treatment and harvesting time for fresh biomass yield, dry matter yield, NDF, ADF, and forage quality parameters, but it had a substantial impact on the ash and mineral compositions. Fresh biomass, DM, WSC, CP, P, and EE were unaffected by the treatment-by-year interaction. Moreover, the interaction of the harvesting stage by year has a greater impact on fresh biomass production, dry matter yield, CF, ash, and Na. Additionally, CF, ash, Na, K, and Mg were considerably impacted by the interaction of treatment, harvesting stage, and year, although CP was slightly affected.

The fresh biomass yield and dry matter yield that were recorded throughout the growing seasons of 2020–2021 and 2021–2022 are shown in Table 4. The result revealed that there were substantial differences between the treatments, with mixed cropping of cereals and alfalfa having a notable effect on the fresh biomass yield and dry matter yield in both cropping seasons. Similar to our study, numerous studies have also highlighted the advantages of alfalfa in a mixed cropping system. Using alfalfa in mixed crops has had a variety of effects. Grabber [28] discovered that intercropping alfalfa and maize increases maize biomass yield. Intercropping alfalfa with grass efficiently enhances fodder DMY in grazing areas, claim Amaraei et al. [29]. Alfalfa and wheat intercropping also considerably increase wheat productivity by improving weed control [30]. Additionally, Be’langer et al. [31] found that intercropping alfalfa with Timothy had a similar effect on the DMY of Timothy as compared to solitary planting. Grass–alfalfa mixes produced superior results than solo cropping on the alfalfa, orchard grass, and tall fescue monocultures in British Columbia [32]. Contrarily, lower grain and biomass yields were found under alfalfa intercropping with corn, according to Berti et al. [33], because of the intercropped alfalfa’s competition for water, which is a crucial element for maize growth.

Additionally, the highest values of dry matter yield were recorded in mixtures of oats or barley with peas in the study of cereal/legume intercropping [20]. However, in this current study, the cereal–pea mixed cropping received a lower FBY and DMY values than the sole cropping of cereals and cereal–alfalfa mixed cropping. Similar to this, Sultan et al. assessed the lower forage yield and dry matter yield in the canola–pea blend [34]. According to an optical review, cereal appears to emerge and establish itself more quickly than peas, which may enable it to grow vigorously below and above ground and more efficiently use resources (nutrients, light, and water) than peas. Early plant emergence and faster initial growth were observed to boost one species’ dominance over another in terms of competition and resource acquisition, resulting in higher biomass growth and yield [35].

Furthermore, the fresh biomass yield and dry matter yield were similarly impacted by harvesting time and had a higher value when harvested at the soft dough stage, although there were no appreciable variations between the cropping seasons. On triticale grown in Northern Italy, Francia et al. [36] predicted and supported the enhanced yield of DMY at the soft dough stage. Similar to this, an earlier study found that the soft dough stage should be harvested for pasture with higher yields of dry matter [15].

Annual ryegrass is a widely used fodder grass that is broadly farmed throughout Europe, America, and Asia. Due to its exceptional forage quality, it has been developed as the primary feed source for herbivorous animals, especially in the winter. Furthermore, ryegrass is frequently employed in crop rotation and intercropping systems with legume species because it will enable forage to have an advanced nutritional concentration [37]. Ryegrass has emerged as one of the cereal crops in the current study with the highest fresh biomass output and dry matter yield due to the following exceptional performance in forage production. Additionally, the current study found that in terms of FBY and DMY, the wheat cultivar Baomai 9 performed better than the Xiaoyan 17 variety. It might be a result of Baomai 9’s increased plant height and better tillering performance.

### 3.2. Nutrient Compositions

The key component parameters for forage production, including dry matter (DM%), crude protein (CP%), water-soluble carbohydrate (WSC%), crude fat (ether extract, EE%), ash, crude fiber (CF%), neutral detergent fiber (NDF%), and acid detergent fiber (ADF%), were measured in the growing seasons of 2020–2021 and 2021–2022. The outcomes are shown in Table 5a,b for both growing seasons. The results determined that treatment W2A (wheat cultivar Xiaoyan 17 and alfalfa mixed cropping) had the highest DM% in both growing seasons, 88.6 and 89.40, respectively. The lowest DM% was found in RP, ryegrass-pea mixed cropping (27.29) in the first growing season, and W1P, Xiaoyan 17 wheat cultivar, and pea mixed cropping (26.06) in the second growing season. In the current study, mixed cropping of cereal–legume (alfalfa) significantly affected the nutrient compositions. According to Lithourgiis et al. [10], intercropping cereals and legumes have various advantages over monocultures, including higher DM, improved land-use efficiency and crop production stability, superior consumption of light, and nutrients, and improved soil conservation. Compared to cereals alone, annual legume–cereal mixes often produce high yields and high-quality features [38]. In comparison to only cereal crops, the most significant profits of legume–cereal combinations have been increased by CP yield [20].

As previously noted, nutrient concentrations were advanced in mixed cropping with alfalfa [15]. Ryegrass–alfalfa mixed cropping was shown to have higher values of WSC%, CP%, EE%, CF%, and ash% in both growing seasons. The values were WSC% (15.82%), CP% (10.78%), EE% (2.30%), CF% (32.06%), and ash% (10.68%) for 2020–2021 cropping seasons and WSC% (15.03%), CP% (11.68%), EE% (3.30%), CF% (32.92%), and ash% (11.07%) for 2021–2022 cropping seasons, respectively. The lowest nutritional compositions were found in wheat cultivar Xiaoyan 17 and pea mixed cropping, with WSC% (4.53%), CP% (6.32%), EE% (0.25%), CF% (17.55%), and ash% (5.56%) for the 2020–2021 cropping seasons and WSC% (3.60%), CP% (6.74%), EE% (0.40%), CF% (19.67%), and ash% (4.37%) for the 2021–2022 cropping seasons, respectively. Likewise, Zhang et al. [39], concluded that cropping combinations of grasses and alfalfa could improve the available nutrient compositions in terms of total biomass output and dietary compositions. When compared to timothy mono-cropping, the alfalfa–timothy mixture’s CP concentration was, however, lower [32]. In contrast to our investigation, the alfalfa–tall fescue blend study [33] found reduced necessary nutritive components, especially CP.

Additionally, the nutritional contents of forage were negatively impacted by the addition of peas to cereal in the present study. Similar to the current study, Sultan et al. evaluated that the greatest nutritional values were exhibited in monocrop pea and canola when compared to mixed cropping treatments [34]. Overall, the results of this investigation are consistent with those of earlier studies [28] due to the lack of substantial changes between mixed cropping regimens involving the legumes pea and the cereals oat, barley, and triticale. On the other hand, Monica et al., assessed the increased protein content in the study of the mixed oat/pea cropping compared to the cereal oat mono-farming [20]. Giacomini et al. [38] also came to the conclusion that triticale produced more crude protein when combined with vetch and pea.

On the other hand, NDF and ADF concentrations have been negatively affected by cereal–alfalfa mixed cropping in this research. The advanced NDF and ADF concentrations were observed in Baomai 9 and pea mixing (57.28 and 33.74 for 2020–2021 and 60.02 and 34.1 for 2021–2022) and the lowest values were in Xiaoyan 17 and alfalfa mixed cropping (34.76 and 17.33 for 2020–2021 and 38.91 and 17.54 for 2021–2022). Similar to this, Contreras-Gova et al. [16] determined that intercropping wheat and clover improved the quality of the fodder in terms of NDF and ADF concentration when compared to solely planting wheat. This might be due to the increased proportion of grass in the mixture; as a result, grasses have more fiber than species of legumes. Sleugh et al. [40] observed a 30% decrease in NDF levels in Kura clover-wheat grass intercropping compared to solo cropping, which is a comparable finding in the current study. Additionally, Kunelius et al. [41] discovered that among grass–legumes mixed cropping, combinations comprising red clover or alfalfa had lower NDF levels.

Furthermore, the harvesting stage had an impact on WSC%, CP%, EE%, CF%, ash%, NDF%, and ADF%, whereas increased nutritional compositions were seen in the soft dough stage compared with the maturity stage, although a larger dry matter content was seen in the maturity stage compared to the soft dough stage based on the development stages of cereal. The yield and quality of semi-leafless grain peas (Pisum sativum L.) were studied by Borreani et al. [42], who discovered that DM increased as maturity advanced, a similar result to our study. Some studies reported that crop CP content and maturity-related declines were consistent [43]. Similarly, as the wheat forage matured, CP decreased, according to other studies that looked at the impact of harvest time [44]. John et al. [45] also concluded that cereals’ early growth phases have a higher CP content than their later growth stages in cereal and cereal/vetch crops for fodder conversation. On the other hand, as the crops matured, it was found that the WSC content of both cereal and cereal/vetch crops was dropping [45]. We primarily ascribe this to a decrease in WSC content in cereal crops as a result of grain growth. In addition, there is an association between ADF and NDF content and cereal crop maturity, with the two variables first rising from the boot stage and later falling with grain filling [46]. In the same way as our result, John et al. concluded that the soft/mid-dough stage was when the NDF and ADF concentration of cereals was highest [45].

Moreover, cropping seasons had no visible influence on the nutritional composition of forage. The wheat cultivar Xiaoyan 17 had the lowest values when the cultivars were compared for their nutritional content of ryegrass crops. The Baomai 9 wheat cultivar was advised for forage production in contrast to the Xiaoyan 17 wheat cultivar. In contrast to Xiaoyan 17, the earlier study found that Baomai 9 had similar results, which were recommended for fodder production. It is possible that compared to the wheat cultivar Xiaoyan 17, Baiomai 9 has higher physiological traits such as a higher tillering capacity and greater plant height.

### 3.3. Scatterplot Matrix Analysis of the Forage of Cereal Mono-Cropping and Cereal−Legume Mixed Cropping

In order to observe and understand the correlation between various variables of WSC, CP, EE, CF, ash, NDF, and ADF of cereal monocropping and cereal–legume mixed cropping, Figure 2 describes the scatterplot matrix analysis of fresh forage of nutritional parameters. All nutritional indicators were positively correlated with each other. While WSC, EE, CF, ash, NDF, and ADF were substantially positively connected with one another, CP slightly interacted with WSC, EE, CF, ash, NDF, and ADF. Similar to the previous study’s findings, NDF and ADF were found to have a positive connection with CP. On the other hand, the association between CP, CF, and NDF was reported to be negative by Soe Htet et al. [14] and Chaudhary et al. [47]. Conversely, Be’langer et al. [31] proposed an enhanced connection between nutritional concentrations with no negative effects on pasture quality by combining timothy and alfalfa.

### 3.4. Mineral Compositions, and Forage Quality Analysis

Table 6 shows the mineral compositions for Ca, Na, K, P, and Mg during the two growing seasons. The findings showed that there were slight variations among the treatments, with higher mineral compositions recorded in RA, ryegrass–alfalfa mixed cropping, Ca (0.37), Na (0.20), K (2.32), P (0.27), and Mg (0.46) for the first growing season, and Ca (0.35), Na (0.19), K (2.02), P (0.31), and Mg (0.47) for the second growing season. Lower mineral compositions were typically observed in mixed cropping of the wheat variety Xiaoyan 17 and pea. Furthermore, there were no significant changes among the cropping seasons, although the study examined the higher mineral content in the soft dough stage compared to those harvested at the maturity stage. When different cultivars were evaluated in this study, ryegrass was found to be suitable for the production of forage in terms of mineral compositions, and the Baomai 9 wheat variety was supported to acquire greater mineral compositions than the Xioayan 17 variety. Similar to the previous study, the ryegrass–alfalfa mixed crop was harvested at the flowering stage and had high mineral compositions (Ca, K, P, and Mg), with the exception of Na [15]. According to Pirhofer-Walzl et al. [48], grass-legume herb combinations showed greater mineral compositions when compared to grass monocultures in terms of K, Mg, Ca, Mn, and Fe. While the mineral compositions of legumes consistently persisted, various mineral concentrations in the blend of herbs and grasses during the early harvesting stage dramatically increased in comparison to the current data [48].

The following relative forage quality parameters: dry matter intake (DMI), digestible dry matter (DDM), relative feed value (RFV), total digestible nutrient (TDN), relative forage quality (RFQ), and quality index (QI) were expressed in Figure 3. Forage quality analysis was considerably impacted by cereal–legume mixed cropping; however, the highest forage quality values; DMI, DDM, RFV, TDN, RFQ, and QI, were found in RA (mixed cropping of ryegrass and alfalfa). The W1P, Xiaoyan 17-pea mixed cropping was found to have inferior fodder quality characteristics. According to the RFV scale, RFV values below 100 are regarded as being lower than the fundamental starting point, which is RFV 100. Sultan et al. also hypothesized that the monocrop pea had the lowest RFV value and that the canola–pea mixed treatment, at a 75:25 planting ratio, had the highest RFV value. They came to the conclusion that the combination planting ratios of the mixed-cropping treatments showed a declining trend in RFV value along with an increase in pea. The RFV value of the cereal–pea mixed cropping was also the lowest among all treatments in the current investigation. DMI is an estimate of how much feed an animal will consume as a percentage of its body weight. Variability in DDM may be explained by variations in CP and cell wall (ADF and NDF) contents [27].

Sultan et al., concluded that TDN was found to be greater in canola–pea mixed-cropping treatments compared to both monocrop pea and monocrop canola treatments, suggesting that combining these two crop species may boost TDN [34]. The TDN forage parameter, which is related to the NDF and ADF concentration of the forage [41], indicates nutrients in the forage that are available to livestock. Higher ADF and NDF values were found in the cereal monocrop and cereal–pea mixed cropping treatments in the current study, which led to lower TDN than in the cereal–alfalfa mixed cropping treatments. According to Lanyasunya et al., the QI and RFQ values of grasses ranged from 1.41 to 1.8 and 105.08 to 138.36 percent, respectively [18]. Moore et al. [27] came to the conclusion that the QI value is less than 1.0 and stated that poor fodder quality and weight loss should be anticipated. The RFV, RFQ, and QI values for solitary Columbus grass (*sorghum almum*) were more significant than mixed cropping with hairy vetch, in contrast to when hairy vetch (*Vicia villosa*) and Columbus grass (*Sorghum almum*) were mixed [27].

According to the higher forage quality values, ryegrass was the cereal crop that was best suited for producing fodder. When comparing the wheat cultivars, wheat cultivar Baomai 9 was recommended because it produced forage that was of higher forage quality values, according to an investigation of forage quality. Additionally, forage harvested at the soft dough stage had greater forage quality characteristics than forage taken at the maturity stage. There were no visible variations in forage quality measured by cropping season. In our study, cereal–alfalfa mixed-cropping treatments generally yielded superior quality forage compared with monocrop cereal and cereal–pea mixed cropping treatments. As a result, cereal–alfalfa mixtures could meet the nutrient requirements for dairy and may provide alternative forage for livestock production.

## 4. Conclusions

In this study, we have determined the impacts of cereal monoculture and cereal–legume mixed cultivation on biomass yield, dry matter yield, nutritional compositions, mineral compositions, and fodder quality index at various harvesting stages in the Loess Plateau region. Among the combinations of the cereal–legume mixture, cereal–alfalfa mixed cropping was the best source of biomass yield and nutritionally rich forage to support the livestock industry. However, peas to cereal had a negative effect on biomass yield, nutritional composition, mineral composition, and forage quality index. Ryegrass has emerged as one of the cereal crops in the current study with the highest fresh biomass output, dry matter yield, nutritional composition, and forage quality index. Moreover, the harvesting at the soft dough stage was recommended for mixed cropping of cereal–legume forage production when compared with the harvesting at the maturity stage. These results can provide valuable information and benefit livestock growers or small-scale farmers growing quality forage in the Loess Plateau region. More studies on cereal and alfalfa mixed cropping are required to investigate various harvesting stages, cultivation techniques, and silage preservation techniques. Additionally, feeding studies are needed to confirm the above-mentioned result in animal performance on the cereal–legume mixed cropping.

## Figures and Tables

**Figure 1 plants-11-02801-f001:**
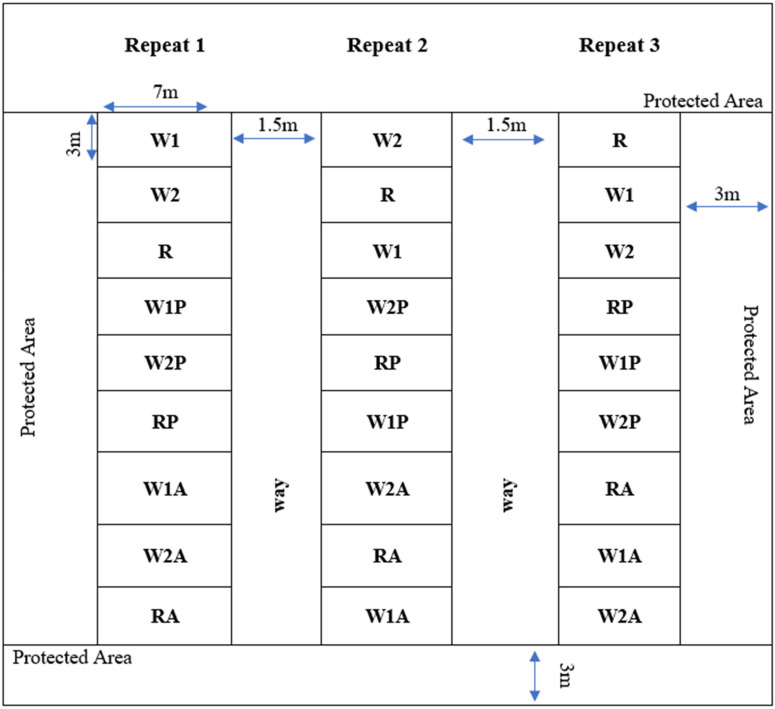
Field layout of cereals mono cropping and cereals–legumes mixed cropping in the experiment.

**Figure 2 plants-11-02801-f002:**
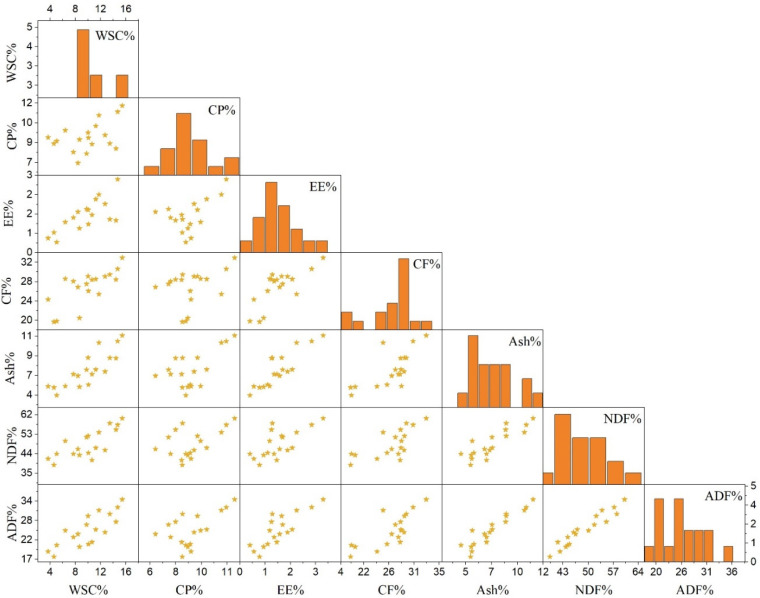
Scattered plot matrix analyses between water-soluble carbohydrates (WSC), crude protein (CP), ether extract (EE), crude fiber (CF), ash, neutral detergent fiber (NDF), and acid detergent fiber (ADF) of forage conducted by cereal mono-cropping and cereal–legume mixed cropping. Diagonal boxes showed histograms for each variable. The lower triangular matrix shows the relationship between a pair of variables.

**Figure 3 plants-11-02801-f003:**
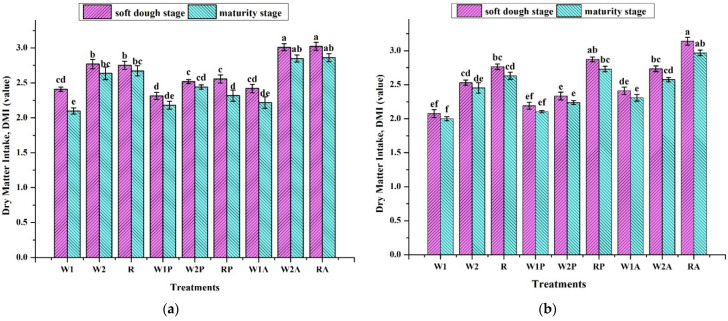
(**a**) Dry matter intake for 2020 growing season, (**b**) dry matter intake for 2021 growing season, (**c**) digestible dry matter for 2020 growing season, (**d**) digestible dry matter for 2021 growing season, (**e**) relative feed value for 2020 growing season, (**f**) relative feed value for 2021 growing season, (**g**) total digestible nutrient for 2020 growing season, (**h**) total digestible nutrient for 2021 growing season, (**i**) quality index for 2020 growing season, (**j**) quality index for 2021 growing season, (**k**) relative forage quality for 2020 growing season, (**l**) relative forage quality for 2021 growing season; treatments express as W1: sole cropping of Xiaoyan 17 wheat cultivar, W2: sole cropping of Biaomai 9 wheat cultivar, R: mono-cropping ryegrass cultivar, W1P: Xiaoyan 17 and pea mixed cropping, W2P: Biaomai 9 and pea mixed cropping, RP: ryegrass and pea mixed cropping, W1A: Xiaoyan 17 and alfalfa mixed cropping, W2A: Biaomai 9 and alfalfa mixed cropping, and RA: ryegrass and alfalfa mixed cropping. The error bars are the standard error of the mean. Different letters mean there are significant differences between the treatments at *p* < 0.05.

**Table 1 plants-11-02801-t001:** Meteorological data during the cropping seasons (2020–2021 and 2021–2022).

Month	Monthly Precipitation (mm)		Monthly Average Temperature (°C)	
	2020–2021	2021–2022	2020–2021	2021–2022
October	19.3	16.96	17.31	16.38
November	19.47	13.34	6.85	5.75
December	0.41	0.56	2.56	1.65
January	0.73	0.64	8.34	1.39
February	2.65	6.96	14.30	3.5
March	6.74	9.06	16.12	13.28
April	42.59	46.68	18.68	17.48
May	34.3	44.46	27.52	27.71

**Table 2 plants-11-02801-t002:** Chemical and physical properties of soil.

Parameters	Value
Total Nitrogen (N)	1.56 g kg^−1^
Phosphorus (P)	16.69 mg kg^−1^
Available Potassium (K)	182.2 mg kg^−1^
Organic Matter (OM)	18.02 g kg^−1^
pH	7.9
Fertility	Medium-fertility

**Table 3 plants-11-02801-t003:** (**a**). Analysis of variance on yield parameters, nutritional compositions, mineral compositions, and forage quality parameters during two cropping seasons: 2020–2021 and 2021–2022. (**b**). Analysis of variance on yield parameters, nutritional compositions, mineral compositions, and forage quality parameters during two cropping seasons: 2020–2021 and 2021–2022.

(a)											
Source of Variation	Agronomic Parameters		Nutritional Compositions								
	FBY	DMY	DM	WSC	CP	EE	CF	Ash	NDF	ADF	
Tr	2.76 **	4.14 **	15.7 **	1.67	7.20 **	5.33 **	4.98 **	4.74 **	13.17 **	12.54 **	
HS	9.14 **	0.26	23.02 **	21.83 **	14.14 **	12.90 **	16.40 **	8.51 **	4.24 **	8.36 **	
Y	0.89	1.51	1.30	1.41	1.07	0.02	1.43	0.05	5.79 *	1.13	
Tr*HS	1.94	0.92	14.56 **	4.10 **	8.82 **	12.23 **	18.90 **	12.91 **	0.10	0.80	
Tr*Y	0.850	2.71 **	0.479	1.83	1.69	0.120	18.67 **	11.22 **	12.58 **	23.87 **	
HS*Y	36.94 **	45.26 **	0.45	3.21	2.20	0.11	0.88 **	10.19 **	0.98	0.72	
Tr*HS*Y	0.43	0.34	0.94	1.57	2.47 *	0.93	53.98 **	7.35 **	0.92	0.57	
**(b)**											
**Source of Variation**	**Mineral** **Compositions**					**Forage Quality** **Index**					
	**Ca**	**Na**	**K**	**P**	**Mg**	**DMI**	**DDM**	**RFV**	**TDN**	**RFQ**	**QI**
Tr	13.14 **	15.75 **	19.46 **	4.87 **	15.23 **	6.10 **	5.64 **	7.63 **	12.37 **	7.74 **	7.53 **
HS	4.2 **	0.17	7.01 **	0.91	8.04 **	11.00 **	14.4 **	4.51 **	8.15 **	18.9 **	13.50 **
Y	0.36	6.02 **	0.98	0.45	0.43	5.42 *	5.30 *	1.67	4.51 *	4.24 *	4.37 *
Tr*HS	10.07 **	13.66 **	16.25 **	4.11 **	11.90 **	0.66	0.33	0.67	2.24 *	0.60	0.61
Tr*Y	14.49 **	18.99 **	6.64 **	1.35	15.32 **	7.10 **	23.82 **	8.78 **	12.16 **	7.74 **	7.75 **
HS*Y	2.64	14.16 **	0.50	0.42	0.43	0.92	0.83	0.72	0.57	0.61	0.93
Tr*HS*Y	0.21	12.42 **	4.48 **	1.2	17.10 **	0.03	0.59	0.02	0.50	0.05	0.05

FBY: fresh biomass yield, DMY: dry matter yield, WSC: water-soluble carbohydrate, CP: crude protein, EE: ether extract, CF: crude fiber, ash, NDF: neutral detergent fiber, ADF: acid detergent fiber, values: ns: non-significant, * and **: significant differences at the 0.05 and 0.01 probability levels. Ca: calcium, Na: sodium, K: potassium, P: phosphorus, Mg: magnesium, DMI: dry matter intake, DDM: digestible dry matter, RFV: relative feed value, TDN: total digestible nutrient, RFQ: relative forage quality, QI: quality index, values: F values ns: non-significant, * and **: significant differences at the 0.05 and 0.01 probability levels.

**Table 4 plants-11-02801-t004:** Fresh biomass yield and dry matter yield of forage of cereal mono-cropping and cereal–legume mixed cropping during 2020−2021 and 2021−2022 growing seasons.

Treatment ^a^	Harvesting Stages ^b^	Parameters (%) ^c^			
		FBY	DMY	FBY	DMY
		2020−2021		2021−2022	
**W1**	**SDS**	42.72 b	15.14 ab	38.93 abc	15.19 bc
	**MS**	24.23 e	14.13 bc	25.67 f	14.92 c
**W2**	**SDS**	46.38 ab	15.63 ab	42.27 ab	18.45 ab
	**MS**	24.42 e	14.24 bc	29.07 ef	16.23 b
**R**	**SDS**	46.57 ab	12.38 d	42.73 ab	11.69 de
	**MS**	31.57 cd	11.67 de	34.13 cde	10.89 de
**W1P**	**SDS**	42.35 b	13.33 cd	32.73 cde	13.57 cd
	**MS**	24.05 e	12.59 d	22.93 g	12.85 d
**W2P**	**SDS**	46.20 ab	15.58 ab	37.53 abc	15.73 bc
	**MS**	24.05 e	14.84 bc	26.27 ef	14.88 c
**RP**	**SDS**	44.92 ab	12.66 d	43.87 ab	12.58 d
	**MS**	29.00 cd	11.20 de	27.27 ef	10.57 e
**W1A**	**SDS**	43.83 ab	16.57 ab	38.07 abc	15.46 bc
	**MS**	26.25 de	16.27 ab	23.33 g	15.11 c
**W2A**	**SDS**	49.83 a	18.78 a	48.87 a	19.77 a
	**MS**	27.35 de	16.48 ab	24.87 g	16.34 b
**RA**	**SDS**	51.88 a	14.88 bc	49.67 a	14.56 c
	**MS**	35.05 c	10.40 e	33.07 cde	11.01 de
**SEM**		4.24	1.46	4.4	2.7
**LOS**		**	**	**	*

Values are means of 3 replicates, LOS: level of significance, SEM: standard error of mean, ns, not significant. * and **: significant differences at the 0.05 and 0.01 probability levels, A significant difference is shown by different letters in the same column. ^a^ W1: wheat cultivar 1 (Xiaoyan 17) sole cropping; W2: wheat cultivar 2 (Baomai 9) sole cropping; R: ryegrass sole cropping; W1P: wheat cultivar 1 (Xiaoyan 17) with pea mixed cropping; W2P: wheat cultivar 2 (Baomai 9) with pea mixed cropping; RP: ryegrass with pea mixed cropping, W1A: wheat cultivar 1 (Xiaoyan 17) with alfalfa mixed cropping; W2A: wheat cultivar 2 (Baomai 9) with alfalfa mixed cropping; RA: ryegrass with alfalfa mixed cropping, ^b^ SDS: harvested at soft dough stage of cereal (cereal); MS: harvested at maturity stage of cereal (cereal), ^c^ FBY: fresh biomass yield, DMY: dry matter yield.

**Table 5 plants-11-02801-t005:** (**a**) Effects of cereal–legume mixed cropping on nutrient compositions of fresh forage during 2020–2021 cropping season. (**b**) Effects of cereal–legume mixed cropping on proximate compositions of fresh forage during 2021–2022 cropping season.

(a)									
Treatment ^a^	Harvesting Stages ^b^	Parameters (%) ^c^							
		DM	WSC	CP	EE	CF	Ash	NDF	ADF
		2020−2021							
**W1**	**SDS**	36.64 c	11.38 b	8.49 bc	0.36 e	27.25 bc	7.11 cd	42.04 de	23.69 cd
	**MS**	81.67 b	4.57 d	6.95 d	0.46 de	18.32 e	5.65 e	36.93 e	19.49 de
**W2**	**SDS**	40.19 c	12.00 b	9.10 ab	1.56 c	28.66 ab	7.64 c	45.56 d	25.97 c
	**MS**	84.33 b	6.23 d	7.91 cd	0.89 d	25.25 cd	5.74 e	39.36 de	19.15 de
**R**	**SDS**	28.14 d	15.35 a	9.38 ab	2.10 ab	27.56 bc	10.05 a	51.81 b	31.23 ab
	**MS**	42.71 c	10.03 b	8.60 bc	1.84 bc	25.61 cd	7.77 c	47.61 c	28.85 b
**W1P**	**SDS**	30.30 d	9.00 c	6.83 d	0.29 e	26.02 c	7.27 c	49.13 bc	29.34 b
	**MS**	80.60 b	4.53 d	6.32 d	0.25 e	17.55 e	5.56 e	38.84 e	24.28 c
**W2P**	**SDS**	39.30 c	10.90 b	8.61 bc	1.40 c	27.71 bc	7.48 c	57.28 a	33.74 a
	**MS**	81.07 b	5.00 d	7.49 cd	0.66 de	23.28 d	5.65 e	40.62 de	25.14 c
**RP**	**SDS**	27.29 d	13.29 ab	9.80 ab	1.94 ab	26.06 c	8.62 b	54.12 b	33.05 a
	**MS**	37.38 c	8.87 c	7.37 cd	1.25 c	25.06 cd	7.51 c	49.92 bc	29.54 b
**W1A**	**SDS**	40.64 c	14.68 ab	9.49 ab	0.69 d	28.47 ab	6.76 d	41.96 de	21.53 d
	**MS**	83.27 b	5.30 d	9.17 ab	0.58 de	24.34 d	6.04 d	34.76 f	17.33 e
**W2A**	**SDS**	40.73 c	15.37 a	9.20 ab	2.18 a	29.27 ab	9.91 ab	42.82 de	23.35 cd
	**MS**	88.60 a	9.67 c	8.90 bc	1.07 cd	24.65 d	7.59 c	37.60 e	21.77 d
**RA**	**SDS**	29.36 d	15.82 a	10.78 a	2.30 a	32.06 a	10.68 a	49.80 bc	28.48 b
	**MS**	45.28 c	11.27 b	8.93 bc	1.78 bc	27.05 bc	8.84 b	45.08 d	27.03 bc
**SEM**		2.61	2.86	0.57	0.03	0.33	0.23	0.50	0.37
**LOS**		**	*	*	**	**	**	**	**
**(b)**									
**Treatment ^a^**	**Harvesting Stages ^b^**	**Parameters (%) ^c^**							
		**DM**	**WSC**	**CP**	**EE**	**CF**	**Ash**	**NDF**	**ADF**
		**2021−2022**							
**W1**	**SDS**	38.37 c	8.20 c	8.63 bc	1.46 cd	28.37 bc	6.74 de	41.75 ef	23.22 de
	**MS**	82.47 ab	4.50 d	7.69 cd	0.56 e	19.81 f	5.25 e	41.11 ef	19.02 f
**W2**	**SDS**	39.11 b	10.97 bc	9.14 bc	1.89 cd	29.05 b	7.27 d	46.64 c	25.41 d
	**MS**	85.53 a	8.45 c	8.91 bc	1.10 d	26.05 de	5.41 e	44.44 e	21.21 ef
**R**	**SDS**	39.49 b	14.33 a	11.16 a	2.84 b	30.63 a	10.43 b	53.66 bc	30.98 b
	**MS**	42.70 b	13.13 ab	7.99 cd	1.29 d	28.43 bc	8.52 c	54.87 b	27.65 c
**W1P**	**SDS**	26.06 d	7.50 cd	8.41 c	1.36 d	28.06 bc	6.74 de	46.15 d	24.08 de
	**MS**	80.40 ab	3.60 d	6.74 d	0.40 e	19.67 f	4.37 f	43.95 e	20.88 ef
**W2P**	**SDS**	26.49 d	9.77 bc	9.36 bc	1.66 cd	28.51 bc	7.04 de	60.02 a	34.10 a
	**MS**	81.07 ab	6.25 cd	8.76 bc	0.94 e	20.48 ef	5.32 e	49.81 c	25.11 d
**RP**	**SDS**	34.74 c	11.45 ab	10.85 ab	2.25 c	27.53 cd	10.26 b	57.07 ab	31.85 b
	**MS**	40.45 b	9.53 bc	7.55 cd	1.25 d	25.39 de	7.25 d	57.82 ab	29.90 bc
**W1A**	**SDS**	28.79 d	10.37 bc	8.95 bc	1.58 cd	26.87 de	6.55 de	43.95 e	21.74 e
	**MS**	86.67 a	4.93 d	8.37 c	0.78 e	24.31 e	5.37 e	38.91 f	17.54 g
**W2A**	**SDS**	37.43 c	12.38 ab	9.93 b	2.07 bc	29.11 b	8.57 c	45.61 de	24.61 de
	**MS**	89.40 a	9.83 bc	9.55 bc	1.18 d	28.58 bc	5.56 e	43.41 e	20.41 e
**RA**	**SDS**	41.58 b	15.03 a	11.68 a	3.30 a	32.92 a	11.07 a	52.01 bc	29.31 bc
	**MS**	44.10 b	14.08 a	8.43 c	1.69 cd	29.45 b	8.56 c	51.46 c	26.78 cd
**SEM**		2.37	1.69	0.40	0.03	0.27	0.18	1.31	0.74
**LOS**		**	**	**	**	**	**	**	**

(**a**): Values are means of 3 replicates, SEM: standard error of mean, LOS: level of significance, ns = not significant. * and **: significant differences at the 0.05 and 0.01 probability levels, A significant difference is shown by different letters in the same column. ^a^ W1: wheat cultivar 1 (Xiaoyan 17) sole cropping; W2: wheat cultivar 2 (Baomai 9) sole cropping; R: ryegrass sole cropping; W1P: wheat cultivar 1 (Xiaoyan 17) with pea mixed cropping; W2P: wheat cultivar 2 (Baomai 9) with pea mixed cropping; RP: ryegrass with pea mixed cropping, W1A: wheat cultivar 1 (Xiaoyan 17) with alfalfa mixed cropping; W2A: wheat cultivar 2 (Baomai 9) with alfalfa mixed cropping; RA: ryegrass with alfalfa mixed cropping, ^b^ SDS: harvested at soft dough stage of cereal (cereal); MS: harvested at maturity stage of cereal (cereal), ^c^ DM: dry matter, WSC: water-soluble carbohydrates, CP: crude protein, EE: ether extract, CF: crude fiber, ash, NDF: neutral detergent fiber, ADF: acid detergent fiber. (**b**): Values are means of 3 replicates, SEM: standard error of mean, LOS: level of significance, ns, not significant. * and **: significant differences at the 0.05 and 0.01 probability levels, A significant difference is shown by different letters in the same column. ^a^ W1: wheat cultivar 1 (Xiaoyan 17) sole cropping; W2: wheat cultivar 2 (Baomai 9) sole cropping; R: ryegrass sole cropping; W1P: wheat cultivar 1 (Xiaoyan 17) with pea mixed cropping; W2P: wheat cultivar 2 (Baomai 9) with pea mixed cropping; RP: ryegrass with pea mixed cropping, W1A: wheat cultivar 1 (Xiaoyan 17) with alfalfa mixed cropping; W2A: wheat cultivar 2 (Baomai 9) with alfalfa mixed cropping; RA: ryegrass with alfalfa mixed cropping, ^b^ SDS: harvested at soft dough stage of cereal (cereal); MS: harvested at maturity stage of cereal (cereal), ^c^ DM: dry matter, WSC = water-soluble carbohydrates, CP: crude protein, EE: ether extract, CF: crude fiber, ash, NDF: neutral detergent fiber, ADF: acid detergent fiber.

**Table 6 plants-11-02801-t006:** Mineral compositions of cereal monocropping and cereal–legume mixed cropping.

Treatment ^a^	Harvesting Stages ^b^	Parameters (%) ^c^									
		Ca	Na	K	P	Mg	Ca	Na	K	P	Mg
		2020−2021					2021−2022				
**W1**	**SDS**	0.18 c	0.02 c	1.71 bc	0.23 ab	0.06 c	0.18 d	0.02 c	1.72 c	0.23 ab	0.05 d
	**MS**	0.17 c	0.02 c	1.65 c	0.22 bc	0.05 c	0.16 d	0.02 c	1.58 cd	0.21 bc	0.04 d
**W2**	**SDS**	0.18 c	0.03 c	1.73 bc	0.24 ab	0.08 c	0.18 d	0.03 c	1.81 bc	0.24 ab	0.07 d
	**MS**	0.17 c	0.02 c	1.63 c	0.22 bc	0.05 c	0.17 d	0.03 c	1.73 c	0.24 ab	0.05 d
**R**	**SDS**	0.28 b	0.19 a	1.92 ab	0.25 a	0.43 a	0.27 b	0.16 ab	1.94 a	0.25 a	0.43 a
	**MS**	0.23 bc	0.11 b	1.85 ab	0.22 bc	0.37 a	0.27 b	0.12 b	1.92 a	0.25 a	0.37 b
**W1P**	**SDS**	0.16 c	0.01 c	1.63 c	0.22 bc	0.05 c	0.17 d	0.01 c	1.63 cd	0.22 bc	0.05 d
	**MS**	0.16 c	0.01 c	1.58 c	0.19 c	0.04 c	0.16 d	0.01 c	1.58 cd	0.19 c	0.02 d
**W2P**	**SDS**	0.18 c	0.03 c	1.58 c	0.25 a	0.07 c	0.17 d	0.02 c	1.65 cd	0.22 bc	0.07 d
	**MS**	0.17 c	0.01 c	0.98 d	0.20 c	0.02 c	0.17 d	0.01 c	1.63 cd	0.20 bc	0.04 d
**RP**	**SDS**	0.26 b	0.12 b	1.81 ab	0.24 ab	0.24 b	0.38 a	0.20 a	0.98 d	0.23 bc	0.24 c
	**MS**	0.23 bc	0.10 b	1.80 ab	0.23 ab	0.22 b	0.35 a	0.11 b	1.80 bc	0.14 d	0.22 c
**W1A**	**SDS**	0.18 c	0.02 c	1.72 bc	0.24 ab	0.06 c	0.19 c	0.03 c	1.85 bc	0.25 ab	0.06 d
	**MS**	0.17 c	0.02 c	1.65 c	0.23 ab	0.05 c	0.17 d	0.02 c	1.65 cd	0.22 ab	0.05 d
**W2A**	**SDS**	0.19 c	0.03 c	1.91 ab	0.25 a	0.08 c	0.20 c	0.04 c	1.93 a	0.26 ab	0.08 d
	**MS**	0.17 c	0.03 c	1.65 c	0.25 a	0.07 c	0.18 d	0.04 c	1.83 bc	0.25 ab	0.07 d
**RA**	**SDS**	0.37 a	0.20 a	2.32 a	0.27 a	0.46 a	0.35 a	0.19 a	2.02 a	0.31 a	0.47 a
	**MS**	0.35 a	0.14 b	2.20 a	0.25 a	0.39 a	0.32 a	0.14 ab	1.95 a	0.24 ab	0.38 b
**SEM**		0.001	0.008	0.05	0.01	0.001	0.006	0.001	0.04	0.01	0.001
**LOS**		**	**	**	*	**	**	**	**	**	**

Values are means of 3 replicates, SEM = standard error of mean, LOS: level of significance, ns: not significant. * and **: significant differences at the 0.05 and 0.01 probability levels. ^a^ W1: wheat cultivar 1 (Xiaoyan 17) sole cropping; W2: wheat cultivar 2 (Baomai 9) sole cropping; R: ryegrass sole cropping; W1P: wheat cultivar 1 (Xiaoyan 17) with pea mixed cropping; W2P: wheat cultivar 2 (Baomai 9) with pea mixed cropping; R P: ryegrass with pea mixed cropping, W1A: wheat cultivar 1 (Xiaoyan 17) with alfalfa mixed cropping; W2A: wheat cultivar 2 (Baomai 9) with alfalfa mixed cropping; RA: ryegrass with alfalfa mixed cropping, ^b^ SDS: harvested at soft dough stage of cereal (cereal); MS: harvested at maturity stage of cereal (cereal), ^c^ Ca: calcium; Na: sodium; K: potassium; P: phosphorus; Mg: magnesium.

## Data Availability

All obtained data are enclosed with this manuscript.

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
