# Peer review of "Influence of Different Harvesting Stages and Cereals–Legume Mixture on Forage Biomass Yield, Nutritional Compositions, and Quality under Loess Plateau Region"

_plants, 2022, doi:10.3390/plants11202801_

Round 1
Reviewer 1 Report
In general
This manuscript contains a critical ambiguity in the design of the research section and in the data analysis section. If the design of the research contains terms from a split-unit design structure, then the data analysis section contains term from single-unit design structure. This manuscript may contain sacrificial pseudo-replication, i.e., the evaluation units were treated as experimental units, which in turn may lead to erroneous statistical inferences.
This manuscript can be accepted after major revisions.
In specific
Lines 170-172. It was written “Three replicates of the split-plot design in a randomized complete block design were used. Harvesting time and mono-cropped and mixed-cropped forages were the main plot treatments and subplots.
Definitions should be given for the following terms: “three replicates”, “split-plot design”, “main plot treatments”, and “subplots”. In this context, authors should use the following statistical terms and their definitions for the “experimental (whole) unit”, the “evaluated unit”, and “degrees of freedom”. After that, authors must decide what type of design should be used (i.e., a single-unit design or/and a split-unit design structure). The following articles: (1-4) should be used to describe the study design section. Physical dimensions between units should be shown in the figure. Is there physical contact between experimental units at the end of the 2020-2021 and 2021-2022 seasons?
1. Hurlbert SH. Pseudofactorialism, response structures and collective responsibility. Austral Ecology, 2013; 38: 646‐663
2. Federer W. T. (1975) The misunderstood split plot. In: Applied Statistics (ed. R. P. Gupta) pp. 9–39. North Holland Publishing, Amsterdam.
3. Hurlbert SH. Pseudoreplication and the design of ecological field experiments. Ecol Monogr, 1984; 54: 187–211
4. Federer W.T. (1977) Sampling, blocking and modeling considerations for split plot and split block designs. Biom. J. 19, 181–200.
Line 262. ANOVA model, sample sizes, and degrees of freedom should be presented in the data analysis section. In addition, the effect size should also be used in this manuscript.
Tables 3a and 3b should show measurement values, effect size values, and exact p-values.
Figure 2 should be converted into table format, sample size, effect sizes, and exact p-values.
Author Response
Response to Reviewer 1 Comments
Comments and Suggestions for Authors
Point 1: Lines 170-172. It was written “Three replicates of the split-plot design in a randomized complete block design were used. Harvesting time and mono-cropped and mixed-cropped forages were the main plot treatments and subplots. Definitions should be given for the following terms: “three replicates”, “split-plot design”, “main plot treatments”, and “subplots”. In this context, authors should use the following statistical terms and their definitions for the “experimental (whole) unit”, the “evaluated unit”, and “degrees of freedom”. Physical dimensions between units should be shown in the figure.
Response 1: Thanks for the suggestion. The sentence has been modified in the revised manuscript.
Point 2: Is there physical contact between experimental units at the end of the 2020-2021 and 2021-2022 seasons?
Response 2: There was no physical contact between experimental units at the end of the 2020-2021 and 2021-2022 seasons.
Point 3: Line 262. ANOVA model, sample sizes, and degrees of freedom should be presented in the data analysis section. In addition, the effect size should also be used in this manuscript.
Response 3: Thanks for the suggestion. The sentence has been modified in the revised manuscript.
Point 4: Tables 3a and 3b should show measurement values, effect size values, and exact p-values.
Response 4: Thanks for the suggestion. The sentence has been modified in the revised manuscript.
Point 4: Figure 2 should be converted into table format, sample size, effect sizes, and exact p-values.
Response 4: Thanks for the suggestion. The table has been modified in the revised manuscript. Furthermore, the effect sizes were generally not showed in the tables of the manuscript. Therefore, the effect size values were expressed detail in the following table although there was no expression in the tables of the manuscript.
|
Treatmenta |
Harvesting Stagesb |
Parameters (%)c |
|||
|
FBY |
DMY |
FBY |
DMY |
||
|
2020−2021 |
2021−2022 |
||||
|
W1 |
SDS |
42.72±3.93b |
15.14±3.28ab |
38.93±4.51abc |
15.19±0.39bc |
|
MS |
24.23±2.77e |
14.13±2.44bc |
25.67±3.87f |
14.92±1.76c |
|
|
W2 |
SDS |
46.38±2.65ab |
15.63±2.99ab |
42.27±5.71ab |
18.45±0.35ab |
|
MS |
24.42±1.32e |
14.24±3.81bc |
29.07±3.41ef |
16.23±2.18b |
|
|
R |
SDS |
46.57±2.00ab |
12.38±1.65d |
42.73±3.45ab |
11.69±2.69de |
|
MS |
31.57±3.93cd |
11.67±2.70de |
34.13±3.33cde |
10.89±2.36de |
|
|
W1P |
SDS |
42.35±3.66b |
13.33±2.55cd |
32.73±4.55cde |
13.57±0.21cd |
|
MS |
24.05±1.46e |
12.59±2.82d |
22.93±1.29g |
12.85±0.69d |
|
|
W2P |
SDS |
46.20±2.89ab |
15.58±1.68ab |
37.53±3.93abc |
15.73±0.23bc |
|
MS |
24.05±1.45e |
14.84±4.35bc |
26.27±1.29ef |
14.88±2.60c |
|
|
RP |
SDS |
44.92±3.72ab |
12.66±2.32d |
43.87±3.86ab |
12.58±2.47d |
|
MS |
29.00±2.89cd |
11.20±4.81de |
27.27±2.83ef |
10.57±1.96e |
|
|
W1A |
SDS |
43.83±1.71ab |
16.57±2.99ab |
38.07±3.99abc |
15.46±0.31bc |
|
MS |
26.25±1.91de |
16.27±4.73ab |
23.33±3.18g |
15.11±1.08c |
|
|
W2A |
SDS |
49.83±2.96a |
18.78±2.87a |
48.87±5.88a |
19.77±0.23a |
|
MS |
27.35±1.2de |
16.48±2.99ab |
24.87±3.01g |
16.34±1.76b |
|
|
RA |
SDS |
51.88±2.01a |
14.88±2.64bc |
49.67±2.20a |
14.56±1.46c |
|
MS |
35.05±3.15c |
10.40±3.50e |
33.07±3.52cde |
11.01±1.40de |
|
|
LOS |
|
** |
** |
** |
** |

Reviewer 2 Report
Generally, the manuscript was prepared correctly. Methodology and analysis of results rather don't raise any objections. However, the obtained results do not contribute significantly to the existing state of knowledge. The paper remains rather a listing of results than an integration of the information. An assessment putting the findings into perspective and make a solid conclusion is missing. The authors should emphasize more the novelty and usefulness of the results.
Author Response
Comments and Suggestions for Authors
Point 1: Methodology and analysis of results rather don't raise any objections.
Response 1: Thanks for the suggestion. The sentence has been modified in the revised manuscript.
Point 2: The obtained results do not contribute significantly to the existing state of knowledge.
Response 2: Thanks for the suggestion. The sentence has been modified in the revised manuscript.
Point 3: The paper remains rather a listing of results than an integration of the information.
Response 3: Thanks for the suggestion. The sentence has been modified in the revised manuscript.
Point 4: An assessment putting the findings into perspective and make a solid conclusion is missing.
Response 4: Thanks for the suggestion. The sentence has been modified in the revised manuscript.
Point 5: The authors should emphasize more the novelty and usefulness of the results.
Response 5: Thanks for the suggestion. The sentence has been modified in the revised manuscript.

Round 2
Reviewer 1 Report
In general
Ignoring the comments of the reviewer is a dead-end way to improve the quality of a scientific article. If the authors have a different point of view, then this point of view should be stated in the text of the manuscript. The key point of this manuscript is the design of the experiment; in particular, it is necessary to show how the experiment was carried out. In this case, the degrees of freedom (df) df1 and df2 from the ANOVA table must be presented in the results section. The exact indication of df1 and df2 from the ANOVA table will allow for substantive discussion. Below is a list of recommended reading.
This manuscript can be accepted after major revisions
In specific
2.1. Experimental Site, Treatments, and Design
Lines 170-171. It was written “A randomized block experimental design with 9 and three replications was applied in the experiment (Figure 1)”.
Figure shows only nine blocks. However, the random procedure is not shown. The figure also does not show three replications of nine plots. It is not clear from the text of the manuscript where and when the three repetitions were made.
The following lines, namely line 177 and lines 184-185, do not clarify the experimental design.
Line 177. It was written “Each plot was split for different harvesting times”.
Lines 184-185. “All cultivars were planted on October 8, 2020, and October 29, 2021, during the growing seasons”.
2.4. Data Analysis
Line 250. Authors used “one way ANOVA”. In this context, all three replications must be made at the same time and the correct name for the test is one way, balanced ANOVA. If replications were performed sequentially, then one-way ANOVA was used incorrectly. In this context, repeated measures ANOVA should be used. In this context, different post-hoc test should be used.
3. Results and Discussion
If the authors used the ANOVA test and Duncan's post hoc test, then the following requirements must be met: mean and SD for each group, F ratio with degrees of freedom (Fdf1,df2 = exact value), exact p-value for ANOVA test, and exact p-values for Duncan's post hoc test.
References
1. Hurlbert, S.H. Pseudoreplication and the Design of Ecological Field Experiments. Ecol. Monogr. 1984, 54(2), 187–211.
2. Hurlbert, S.H. Pseudofactorialism, response structures and collective responsibility. Austral Ecology 2013, 38(6), 646-663
3. Hurlbert, S.H.; Lombardi, C.M. Final collapse of the Neyman-Pearson decision theoretic framework and rise of the neoFisherian. Annales Zoologici Fennici. 2009, 46(5), 311-349.
Author Response
Response to Reviewer 1 Comments (Round 2)
Comments and Suggestions for Authors
Point 1: Lines 170-171. It was written “A randomized block experimental design with 9 and three replications was applied in the experiment (Figure 1)”. Figure shows only nine blocks. However, the random procedure is not shown. The figure also does not show three replications of nine plots. It is not clear from the text of the manuscript where and when the three repetitions were made. The following lines, namely line 177 and lines 184-185, do not clarify the experimental design.
Line 177. It was written “Each plot was split for different harvesting times”.
Lines 184-185. “All cultivars were planted on October 8, 2020, and October 29, 2021, during the growing seasons”.
Response 1: Thanks for the suggestion. The figure of the experimental layout has been modified in the revised manuscript.
Point 2: Line 250. Authors used “one way ANOVA”. In this context, all three replications must be made at the same time and the correct name for the test is one-way, balanced ANOVA. If replications were performed sequentially, then one-way ANOVA was used incorrectly. In this context, repeated measures ANOVA should be used. In this context, different post-hoc test should be used.
Response 2: Thanks for the suggestion. I used the univariate analysis in SPSS for mean data and Dunn's Multiple Comparison Test for post-hoc test. The sentence has been modified in the revised manuscript.
Point 3: If the authors used the ANOVA test and Duncan's post hoc test, then the following requirements must be met: mean and SD for each group, F ratio with degrees of freedom (Fdf1,df2 = exact value), exact p-value for ANOVA test, and exact p-values for Duncan's post hoc test.
Response 3: Thanks for the suggestion. The answer of this question has been responded the above.

Round 3
Reviewer 1 Report
Please see the attached file.

Author Response
Response to Reviewer 1 Comments
Comments and Suggestions for Authors
Comment; In general, I currently have three versions of the manuscript: (v1 [plants-1940588-peer-review-v1]), (v2 [plants-1940588-peer-review-v2]) and (v3 [plants-1940588-peer-review-v3]). All three versions (v1-v3) have a different design. This is an amazing phenomenon. I assume that the authors do not assume the following two fundamental properties: a random design and the difference between experimental units and evaluated units. The authors categorically ignore my suggestions to report the sample size and the results of the ANOVA table, in particular the degrees of freedom and the exact values of the ratios of F and p-values when using the ANOVA test and exact p-values for Duncan's post-hoc test. This information is essential for understanding statistical analysis.
Response: Thanks for the suggestion. We assumed that the field level experimental design had been missed. After reading your suggested article, we recognized that the randomized blocked experimental design was used in this research. The sentence has been modified in the revised manuscript. The following table is univariate test that was used in this manuscript for data analysis.
Comment; From the v3, figure 1 shows a plot with the following dimensions: north-south = 33 m and east-west = 30 m. This plot contains nine experimental units with three replications. However, this is not a random design. It is simple segregation “in which the principle of interspersion can be violated” (Hurlbert 1984, Figure 1). Therefore, each of three replications can be affected from different natural (uncontrolled) factors. This is poor experimental design. It should be remembered the recommendation of the father of modern experimental design R.A. Fisher (1938): “To call in the statistician after the experiment is done may be no more than asking him to perform a postmortem examination: he may be able to say what the experiment died of″.
Response: Thanks for the suggestion. From the v3 figure 1, we supposed that this experimental design is the randomized block design according to the Hurlbert 1984, Figure 1. In this experimental field, there were 9 treatments which are 3 sole cropping of cereals, 3 mixed cropping of cereal- pea, and 3 mixed cropping of cereal- alfalfa.
Comment; From manuscript v3, the ANOVA table should show the next two degrees of freedom (df1 = 8 and df2 = 18). This will be the correct answer. Since the design structure contains nine independent groups (G) and three experimental units in each group and hence the total number of experimental units, N = 27. It is context, df1= G - 1, and df2 = N-G. Each experimental unit contains 18 (3 x 6) evaluated units. Evaluated units are dependent units. It should be noted that any two samples from the first replication will be more similar to each other than the same two samples from different replications. If the evaluated units are treated as experimental units, then this leads to sacrificial pseudoreplication. The result of this sacrificial pseudo-replication is artificially inflated degrees of freedom, giving the illusion of having a more powerful test than the data support (Hurlbert 1984).
Response: Thanks for the suggestion. I have read the review paper of Hurlbert 1984 and two degree of freedom have shown in the table.
